# TOWARDS UNDERSTANDING ENSEMBLE, KNOWLEDGE DISTILLATION AND SELF-DISTILLATION IN DEEP LEARNING

**Zeyuan Allen-Zhu**
Meta FAIR Labs
zeyuanallenzhu@meta.com

**Yuanzhi Li**
Mohamed bin Zayed University of AI
Yuanzhi.Li@mbzuai.ac.ae

## ABSTRACT

We formally study how *ensemble* of deep learning models can improve test accuracy, and how the superior performance of ensemble can be distilled into a single model using *knowledge distillation*. We consider the challenging case where the ensemble is simply an average of the outputs of a few independently trained neural networks with the *same* architecture, trained using the *same* algorithm on the *same* data set, and they only differ by the random seeds used in the initialization.

We show that ensemble/knowledge distillation in *deep learning* works very differently from traditional learning theory (such as boosting or NTKs). We develop a theory showing that when data has a structure we refer to as "multi-view", then ensemble of independently trained neural networks can provably improve test accuracy, and such superior test accuracy can also be provably distilled into a single model. Our result sheds light on how ensemble works in deep learning in a way that is completely different from traditional theorems, and how the "dark knowledge" is hidden in the outputs of the ensemble and can be used in distillation.[1]

## 1 INTRODUCTION

Ensemble (Dietterich, 2000; Hansen & Salamon, 1990; Polikar, 2006) is one of the most powerful techniques in practice to improve the performance of deep learning. By simply averaging the outputs of merely a few (like 3 or 10) independently-trained neural networks of the *same* architecture, using the *same* training method over the *same* training data, it can significantly boost the prediction accuracy over the test set comparing to individual models. The only difference is the randomness used to initialize these networks and/or the randomness during training. Moreover, it is discovered by Hinton et al. (2015) that such superior performance of the ensemble can be transferred into a single model (of the same size as the individual models) using a technique called *knowledge distillation*: that is, simply train a single model to match the output of the ensemble (such as "90% cat + 10% car", also known as *soft labels*) as opposite to the true data labels, over the same training data.

On the theory side, there are lots of works studying the superior performance of ensemble from principled perspectives (see full version for citations). However, most of these works only apply to: (1). Boosting: where the coefficients associated with the combinations of the single models are actually trained, instead of simply taking average; (2). Bootstrapping/Bagging: the training data are different for each single model; (3). Ensemble of models of different types and architectures; or (4). Ensemble of random features or decision trees. To the best of our knowledge, *none of these cited works* apply to the particular type of ensemble that is widely used in deep learning: simply take a *uniform* average of the output of the learners, which are neural networks with the *same* architecture and are trained by stochastic gradient descent (SGD) over the *same* training set. In fact, *very critically, for underline{deep learning} models*:

- TRAINING AVERAGE DOES NOT WORK: if one directly trains to learn an average of individual neural networks initialized by different seeds, the performance is much worse than ensemble.

- KNOWLEDGE DISTILLATION WORKS: the superior performance of ensemble in deep learning can be distilled into a single model (Hinton et al., 2015).

---

[1]Full version of this paper can be found on https://arxiv.org/abs/2012.09816.

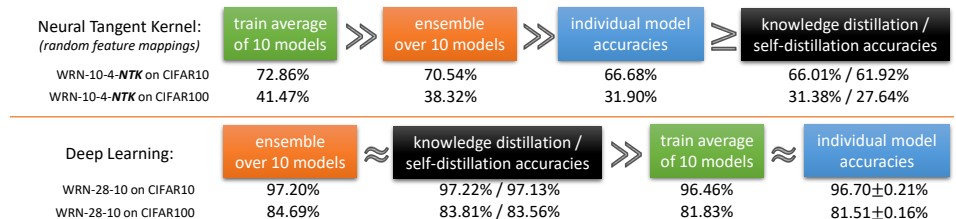

Figure 1: Ensemble in deep learning is very different from ensemble in random feature mappings. Details in Figure 6.

- SELF-DISTILLATION WORKS: even distilling a single model into another of the same size, there is performance boost. (Furlanello et al., 2018; Mobahi et al., 2020; Zhang et al., 2019)

We are unaware of any satisfactory theoretical explanation for the phenomena above. For instance, as we shall argue, some traditional view for why ensemble works, such as 'ensemble can enlarge the feature space in random feature mappings', even give contradictory explanations to the above phenomena, thus *cannot* explain knowledge distillation or ensemble in *deep learning*. Motivated by this gap between theory and practice we study the following question for *multi-class* classification:

---
**Our theoretical questions:**

How does ensemble improve the test-time performance in deep learning when we simply (un-weightedly) average over a few independently trained neural networks? – Especially when all the neural networks have the same architecture, are trained over the same data set using the same standard training algorithm and only differ by the random seeds, and even when all single models already have $100\%$ *training accuracy*? How can such superior test-time performance of ensemble be later "distilled" into a single neural network of the same architecture, simply by training the single model to match the output of the ensemble over the same training data set?

---

**Our results.** We prove for certain multi-class classification tasks with a special structure we refer to as **multi-view**, with a training set $\mathcal{Z}$ consisting of $N$ i.i.d. samples from some unknown distribution $\mathcal{D}$, for certain two-layer convolutional network $f$ with (smoothed-)ReLU activation as learner:

- (Single model has bad test accuracy): there is a value $\mu > 0$ such that when a single model $f$ is trained over $\mathcal{Z}$ using the cross-entropy loss, via gradient descent (GD) starting from random Gaussian initialization, the model can reach zero training error *efficiently*. However, w.h.p. the prediction (classification) error of $f$ over $\mathcal{D}$ is between $0.49\mu$ and $0.51\mu$.

- (Ensemble provably improves test accuracy): let $f_1, f_2, \cdots, f_L$ be $L = \widetilde{\Omega}(1)$ independently trained single models as above, then w.h.p. $G = \frac{1}{L}\sum_\ell f_\ell$ has prediction error $\leq 0.01\mu$ over $\mathcal{D}$.

- (Ensemble can be distilled into a single model): if we further train (using GD from random initialization) another single model $f_0$ (same architecture as each $f_\ell$) to match the output of $G = \frac{1}{L}\sum_\ell f_\ell$ merely over the same training data set $\mathcal{Z}$, then $f_0$ can be trained *efficiently* and w.h.p. $f_0$ will have prediction error $\leq 0.01\mu$ over $\mathcal{D}$ as well.

- (*Self-distillation* also improves test accuracy): if we further train (using GD from random initialization) another single model $f'$ (same architecture as $f_1$) to match the output of *the single model $f_1$* merely over the same training data set $\mathcal{Z}$, then $f'$ can be trained *efficiently* and w.h.p. has prediction error at most $\leq 0.26\mu$ over $\mathcal{D}$. The main idea is that self-distillation is performing "*implicit ensemble + knowledge distillation*", as we shall argue in Section 4.2.

We defer discussions of our empirical results to Section 5. However, we highlight some of the empirical findings, as they shall confirm and justify our theoretical approach studying ensemble and knowledge distillation in deep learning. Specifically, we give empirical evidences showing that:

- Knowledge distillation does not work for random feature mappings; and ensemble in deep learning is very different from ensemble in random feature mappings (see Figure 1).

- Special structures in data (such as the "multi-view" structure we shall introduce) is needed for ensemble of neural networks to work.

- The variance due to label noise or the non-convex landscape of training, in the independently-trained models, may not be connected to the superior performance of ensemble in deep learning.

## 2 OUR METHODOLOGY AND INTUITION

### 2.1 A FAILURE ATTEMPT USING RANDOM FEATURE MAPPINGS

The recent advance in deep learning theory shows that under certain circumstances, neural networks can be treated as a linear function over random feature mappings — see (Allen-Zhu et al., 2019b; Arora et al., 2019b; Daniely et al., 2016; Du et al., 2018b; Jacot et al., 2018; Zou et al., 2018) and the references therein. In particular, the theory shows when $f : \mathbb{R}^{D+d} \to \mathbb{R}$ is a neural network with inputs $x \in \mathbb{R}^d$ and weights $W \in \mathbb{R}^D$, in some cases, $f(W, x)$ can be approximated by:

$$f(W, x) \approx f(W_0, x) + \langle W - W_0, \nabla_W f(W_0, x) \rangle$$

where $W_0$ is the random initialization of the neural network, and $\Phi_{W_0}(x) := \nabla_W f(W_0, x)$ is the neural tangent kernel (NTK) feature mapping. This is known as the NTK approach. If this approximation holds, then training a neural network can be approximated by learning a linear function over random features $\Phi_{W_0}(x)$, which is very theory-friendly.

**Ensemble works for random features / NTK.** Traditional theorems (Alhamdoosh & Wang, 2014; Brown et al., 2005a; Bryll et al., 2003; Tsymbal et al., 2005) suggest that the ensemble of independently trained random feature models can indeed *significantly improve* test-time performance, as it enlarges the feature space from $\Phi_{W_0}(x)$ to $\{\Phi_{W_0^{(i)}}(x)\}_{i \in [L]}$ for $L$ many independently sampled $W_0^{(i)}$. This can be viewed as a feature selection process (Alvarez et al., 2012; Cai et al., 2018; Oliveira et al., 2003; Opitz, 1999; Rokach, 2010), and we have confirmed it for NTK in practice, see Figure 1. However, *can we understand ensemble and knowledge distillation in DL as feature selections using NTK?* Unfortunately, our empirical results provide many counter examples towards those arguments, see discussions below and Figure 1.

**Contradiction 1: training average works even better.** Although ensemble of linear functions over NTK features with different random seeds: $f_i(x) = \langle W^{(i)}, \Phi_{W_0^{(i)}}(x) \rangle$ does improve test accuracy, however, such improvement is mainly due to the use of a larger set of random features, whose combinations contain functions that generalize better. To see this, we observe that an even superior performance (than the ensemble) can simply be obtained by directly training $F(x) = \frac{1}{L}(f_1 + f_2 + \cdots + f_L)$ from random initialization. *In contrast,* recall if $f_i(x)$'s are multi-layer neural networks with different random seeds, then training their average barely gives any better performance comparing to individual networks $f_i$, as now all the $f_i$'s are capable of learning the same set of features.

**Contradiction 2: knowledge distillation does not work.** For NTK feature mappings, we observe that the result obtained by ensemble cannot be distilled at all into individual models, indicating the features selected by ensemble *is not contained* in the feature $\Phi_{W_0^{(i)}}(x)$ of any individual model. In contrast, in actual deep learning, ensemble *does not enlarge feature space*: so an individual neural network is capable of learning the features of the ensemble model.

**In sum,** ensemble in deep learning may be very different from ensemble in random features. It may be more accurate to study ensemble / knowledge distillation in deep learning as a *feature learning process*, instead of a feature selection process. But still, we point out a fundamental difficulty:

---
**Key challenge:**

If a single deep learning model is capable of — through knowledge distillation — learning the features of the ensemble model and achieving better test accuracy comparing to training the single model directly (and the same training accuracy, typically at global optimal of $100\%$), then why the single model *cannot learn* these features directly when we train the model to match the true data labels? What is the **dark knowledge** hidden in the output of ensemble (a.k.a. soft label)[2] comparing to the original hard label?

---

### 2.2 ENSEMBLE IN DEEP LEARNING: A FEATURE LEARNING PROCESS

Before addressing the key challenge, we point out that prior works are very limited with respect to studying neural network training as a feature learning process. Most of the existing works proving that neural networks can learn features only focus on the case *when the input is Gaussian or*

---
[2]For a $k$-class classification problem, the output of a model $g(x)$ is usually $k$-dimensional, and represents a soft-max probability distribution over the $k$ target classes. This is known as the *soft label*.

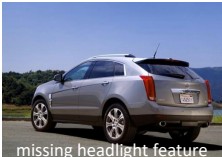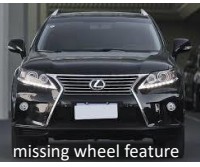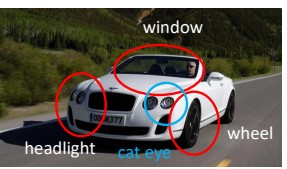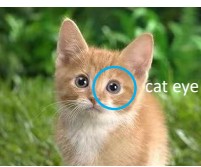

Figure 2: Illustration of images with multiple views (features) in the ImageNet dataset.

*Gaussian-like* — see for instance (Kawaguchi, 2016; Soudry & Carmon, 2016; Xie et al., 2016) and many others. However, as we demonstrate in Figure 7 in the full version,

> **Ensemble in DL might not improve test accuracy when inputs are Gaussian-like:**
>
> Empirically, ensemble *does not* improve test accuracy in deep learning, in certain scenarios when the distribution of the input data is Gaussian or even mixture of Gaussians. This is true over various learner network structures (fully-connected, residual, convolution neural networks) and various labeling functions (when the labels are generated by linear functions, fully-connected, residual, convolutional networks, with/without label noise, with/without classification margin).

**Bias variance view of ensemble:**  Some prior works also try to attribute the benefit of ensemble as reducing the *variance* of individual solutions due to label noise or non-convex landscape of the training objective. However, reducing such variance can reduce a convex test loss (typically cross-entropy), but not necessarily the *test classification error*. Concretely, the synthetic experiments in Figure 7 show that, after applying ensemble over Gaussian-like inputs, the variance of the model outputs is reduced but the test accuracy is *not improved*. We give many more empirical evidences to show that the variance (either from label noise or from the non-convex landscape) is usually not the cause for why ensemble works in deep learning, see Section 5.

Hence, to understand the true benefit of ensemble in deep learning in theory, we would like to study a setting that can *approximate* practical deep learning, where:

- The input distribution is more structured than standard Gaussian and there is no label noise. (From above discussions, ensemble cannot work for deep learning distribution-freely).
- The individual neural networks all are well-trained, in the sense that the training accuracy in the end is $100\%$, and there is nearly no variance in the test accuracy for individual models. (So training never fails.)

In this work, we propose to study a setting of data that we refer to as **multi-view**, where the above two conditions both hold when we train a two-layer neural networks with (smoothed-)ReLU activations. We also argue that the multi-view structure we consider is fairly common in the data sets used in practice, in particular for vision tasks. We give more details below.

## 2.3   OUR APPROACH: LEARNING MULTI-VIEW DATA

Let us first give a thought experiment to illustrate our approach, and we present the precise mathematical definition of the "multi-view" structure in Section 3. Consider a *binary* classification problem and four "features" $v_1, v_2, v_3, v_4$. The first two features correspond to the first class label, and the next two features correspond to the second class label. In the data distribution:

- When the label is class 1, then:[3]

$$\begin{cases} \text{both } v_1, v_2 \text{ appears with weight 1, one of } v_3, v_4 \text{ appears with weight 0.1} & \text{w.p. } 80\%; \\ \text{only } v_1 \text{ appears with weight 1, one of } v_3, v_4 \text{ appears with weight 0.1} & \text{w.p. } 10\%; \\ \text{only } v_2 \text{ appears with weight 1, one of } v_3, v_4 \text{ appears with weight 0.1} & \text{w.p. } 10\%. \end{cases}$$

- When the label is class 2, then

$$\begin{cases} \text{both } v_3, v_4 \text{ appears with weight 1, one of } v_1, v_2 \text{ appears with weight 0.1} & \text{w.p. } 80\%; \\ \text{only } v_3 \text{ appears with weight 1, one of } v_1, v_2 \text{ appears with weight 0.1} & \text{w.p. } 10\%; \\ \text{only } v_4 \text{ appears with weight 1, one of } v_1, v_2 \text{ appears with weight 0.1} & \text{w.p. } 10\%. \end{cases}$$

---

[3]One can for simplicity think of "$v$ appears with weight $\alpha$ and $w$ appears with weight $\beta$" as data $= \alpha v + \beta w + \text{noise}$.

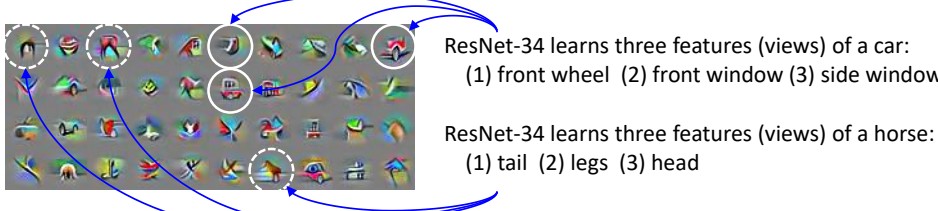

ResNet-34 learns three features (views) of a car:
(1) front wheel  (2) front window (3) side window

ResNet-34 learns three features (views) of a horse:
(1) tail  (2) legs  (3) head

Figure 3: Visualization of the channels in layer-23 of a ResNet-34 trained on CIFAR-10.

We call the $80\%$ of the data *multi-view data*: these are the data where multiple features exist and can be used to classify them correctly. We call the rest $20\%$ of the data *single-view data*: some features for the correct labels are missing. [4]

**How individual neural networks learn.**   Under the multi-view data defined above, if we train a neural network using the cross-entropy loss via gradient descent (GD) from random initialization, during the training process of the individual networks, we show that:

- The network will quickly pick up one of the feature $v \in \{v_1, v_2\}$ for the first label, and one of the features $v' \in \{v_3, v_4\}$ for the second label. So, $90\%$ of the training examples, consisting of all the multi-view data and half of the single-view data (those with feature $v$ or $v'$), are classified correctly. Once classified correctly (with a large margin), these data begin to contribute negligible to gradient by the nature of the cross-entropy loss.

- Next, the network will memorize (using e.g. the noise in the data) the remaining $10\%$ of the training examples without learning any new features, due to insufficient amount of left-over samples after the first phase, thus achieving training accuracy $100\%$ but test accuracy $90\%$.

**How ensemble improves test accuracy.**   It is simple why ensemble works. Depending on the randomness of initialization, each individual network will pick up $v_1$ or $v_2$ each w.p. $50\%$. Hence, as long as we ensemble $\widetilde{O}(1)$ many independently trained models, w.h.p. their ensemble will pick up both features $\{v_1, v_2\}$ and both features $\{v_3, v_4\}$. Thus, all the data will be classified correctly.

**How knowledge distillation works.**   Perhaps less obvious is how knowledge distillation works. Since ensemble learns all the features $v_1, v_2, v_3, v_4$, given a multi-view data with label 1, the ensemble will actually output $\propto (2, 0.1)$, where the 2 comes from features $v_1, v_2$ and $0.1$ comes from one of $v_3, v_4$. On the other hand, an individual model learning only one of $v_3, v_4$ will actually output $\propto (2, 0)$ when the feature $v_3$ or $v_4$ in the data does not match the one learned by the model. Hence, by training the individual model to match the output of the ensemble, the individual model is *forced* to learn both features $v_3, v_4$, even though it has already perfectly classified the training data. **This is the "dark knowledge" hidden in the output of the ensemble model**. (This theoretical finding is consistent with practice: Figure 8 in the full paper suggests that models trained from knowledge distillation should have learned most of the features, and further computing their ensemble does not give much performance boost.)

**Significance of our technique.**   Our work belongs to the generic framework of feature learning in DL where one proves that certain aspects of the algorithm (e.g. the randomness) affects the order

[4]**Meaningfulness of our multi-view hypothesis.**  Such "multi-view" structure is very common in many of the datasets where deep learning excels. In vision datasets in particular, as illustrated in Figure 2, a car image can be classified as a car by looking at the headlights, the wheels, or the windows. For a typical placement of a car in images, we can observe all these features and use any of these features to classify it as a car. However, there are car images taken from a particular angle, where one or more features can be missing. For example, an image of a car facing forward might be missing the wheel feature. Moreover, some car might also have a small fraction of "cat features": for example, the headlight might appear similar to cat eyes the ear of a cat. This can be used as the "dark knowledge" by the single model to learn from the ensemble. In Figure 3, we visualize the learned features from an actual neural network to show that they can indeed capture different views. In Figure 5, we plot the "heatmap" for some car images to illustrate that single models (trained from different random seeds) indeed pick up different parts of the input image to classify it as a car. In Figure 9, we manually delete for instance 7/8 of the channels in some intermediate layer of a ResNet, and show that the test accuracy may not be affected by much after ensemble — thus supporting that the multi-view hypothesis can indeed exist even in the intermediate layers of a neural network and ensemble is indeed collecting all these views.

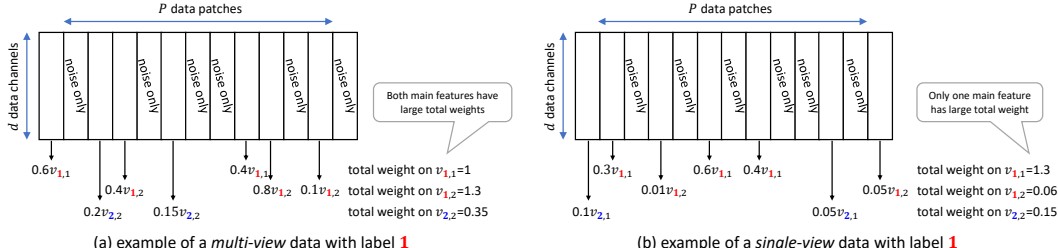

Figure 4: Illustration of a multi-view and a single-view data point; the feature vectors can also be combined with feature noise and random noise, see Def. 3.1.

where features are learned. This is *fundamentally different from convex optimization*, such as kernel method, where (with $\ell_2$ regularization) there is an unique *global minimum* so the choice of the random seed does not matter (thus, ensemble does not help). There are other works that consider other aspects, such as the choice of learning rate, that can affect the order where the features are learned (Li et al., 2019). Our work is fundamentally different: they only focus on the NTK setting where the features are *not learned*; we study a *feature learning* process. Recall, the NTK setting cannot be used to explain ensemble and distillation in DL. Our work extends the reach of traditional machine learning theory, where typically the "generalization" is separated from "optimization." Such "separate" treatment might not be enough to understand how deep learning works.

## 3 PROBLEM SETUP

The "multi-view" data distribution is a straight-forward generalization of the intuitive setting in Section 2.3. For simplicity, in the main body, we use example choices of the parameters mainly a function of $k$ (such as $P = k^2$, $\gamma = \frac{1}{k^{1.5}}$, $\mu = \frac{k^{1.2}}{N}$, $\rho = k^{-0.01}$, $\sigma_0 = 1/\sqrt{k}$ as we shall see), and we consider the case when $k$ is sufficiently large. In our full version, we shall give a much larger range of parameters for the theorems to hold.

### 3.1 DATA DISTRIBUTION AND NOTATIONS

We consider learning a $k$-class classification problem over $P$-patch inputs, where each patch has dimension $d$. In symbols, each labelled data is represented by $(X, y)$ where $X = (x_1, x_2, \cdots, x_P) \in (\mathbb{R}^d)^P$ is the data vector and $y \in [k]$ is the data label. For simplicity, we focus on the case when $P = k^2$, and $d = \text{poly}(k)$ for a large polynomial.

We consider the setting when $k$ is sufficiently large.[5] We use "w.h.p." to denote with probability at least $1 - e^{-\Omega(\log^2 k)}$, and use $\widetilde{O}, \widetilde{\Theta}, \widetilde{\Omega}$ notions to hide polylogarithmic factors in $k$.

We first assume that each label class $j \in [k]$ has multiple associated features, say *two features for the simplicity of math*, represented by unit **feature vectors** $v_{j,1}, v_{j,2} \in \mathbb{R}^d$. For notation simplicity, we assume that all the features are orthogonal, namely,

$$\forall j, j' \in [k], \ \forall \ell, \ell' \in [2], \ \|v_{j,\ell}\|_2 = 1 \quad \text{and} \quad v_{j,\ell} \perp v_{j',\ell'} \text{ when } (j, \ell) \neq (j', \ell')$$

although our work also extends to the "incoherent" case trivially. We denote by

$$\mathcal{V} := \{v_{j,1}, v_{j,2}\}_{j \in [k]} \quad \textit{the set of all features.}$$

We consider the following data and label distribution. Let $C_p$ be a global constant, $s \in [1, k^{0.2}]$ be a sparsity parameter. To be concise, we define the *multi-view distribution* $\mathcal{D}_m$ and *single-view distribution* $\mathcal{D}_s$ together. Due to space limitation, here we hide the specification of the random "noise" and defer it to the full version.[6]

**Definition 3.1** (data distributions $\mathcal{D}_m$ and $\mathcal{D}_s$)**.** *Given $\mathcal{D} \in \{\mathcal{D}_m, \mathcal{D}_s\}$, we define $(X, y) \sim \mathcal{D}$ as follows. First choose the label $y \in [k]$ uniformly at random. Then, the data vector $X$ is generated*

---

[5]If we want to work with fixed $k$, say $k = 2$, our theorem can also be modified to that setting by increasing the number of features per class. We keep our current setting with two features to simplify the notations.

[6]At a high level, we shall allow such "noise" to be any feature noise plus Gaussian noise, such as $noise = \sum_{v' \in \mathcal{V}} \alpha_{p,v'} v' + \xi_p \in \mathbb{R}^d$, where each $\alpha_{p,v'} \in [0, \gamma]$ can be arbitrary, and $\xi_p \sim \mathcal{N}(0, \sigma_p^2 \mathbf{I})$.

*as follows (also illustrated in Figure 4).*

1. *Denote $\mathcal{V}(X) = \{v_{y,1}, v_{y,2}\} \cup \mathcal{V}'$ as the set of feature vectors used in this data vector $X$, where $\mathcal{V}'$ is a set of features uniformly sampled from $\{v_{j',1}, v_{j',2}\}_{j' \in [k] \setminus \{y\}}$, each with probability $\frac{s}{k}$.*

2. *For each $v \in \mathcal{V}(X)$, pick $C_p$ many disjoint patches in $[P]$ and denote it as $\mathcal{P}_v(X) \subset [P]$ (the distribution of these patches can be arbitrary). We denote $\mathcal{P}(X) = \cup_{v \in \mathcal{V}(X)} \mathcal{P}_v(X)$.*

3. *If $\mathcal{D} = \mathcal{D}_s$ is the single-view distribution, pick a value $\widehat{\ell} = \widehat{\ell}(X) \in [2]$ uniformly at random.*

4. *For each $v \in \mathcal{V}(X)$ and $p \in \mathcal{P}_v(X)$, we set $x_p = z_p v + $ "noise" $\in \mathbb{R}^d$, where, the random coefficients $z_p \geq 0$ satisfy that:*

   *In the case of multi-view distribution $\mathcal{D} = \mathcal{D}_m$,*

   - *$\sum_{p \in \mathcal{P}_v(X)} z_p \in [1, O(1)]$ when $v \in \{v_{y,1}, v_{y,2}\}$,  [7]*
   - *$\sum_{p \in \mathcal{P}_v(X)} z_p \in [\Omega(1), 0.4]$ when $v \in \mathcal{V}(X) \setminus \{v_{y,1}, v_{y,2}\}$,  [8]*

   *In the case of single-view distribution $\mathcal{D} = \mathcal{D}_s$,*

   - *$\sum_{p \in \mathcal{P}_v(X)} z_p \in [1, O(1)]$ when $v = v_{y,\widehat{\ell}}$,*
   - *$\sum_{p \in \mathcal{P}_v(X)} z_p \in [\rho, O(\rho)]$ when $v = v_{y,3-\widehat{\ell}}$,*
   - *$\sum_{p \in \mathcal{P}_v(X)} z_p \in [\Omega(\Gamma), \Gamma]$ when $v \in \mathcal{V}(X) \setminus \{v_{y,1}, v_{y,2}\}$.*

5. *For each $p \in [P] \setminus \mathcal{P}(X)$, we set $x_p$ to consist only of "noise".*

*Remark* 3.2. The distribution of how to pick $\mathcal{P}(X)$ and assign $\sum_{p \in \mathcal{P}_v(X)} z_p$ to each patch in $p \in \mathcal{P}_v(X)$ can be arbitrary (and can depend on other randomness in the data as well). In particular, we have allowed **different features** $v_{j,1}, v_{j,2}$ to show up with **different weights** in the data (for example, for multi-view data, some view $v_{y,1}$ can consistently have larger $z_p$ comparing to $v_{y,2}$). Yet, we shall prove that the order to learn these features by the learner network *can still be flipped* depending on the randomness of network initialization.

**Interpretation of our data distribution.** As we argue more in the full paper, our setting can be tied to a down-sized version of convolutional networks applied to image classification data. With a small kernel size, good features in an image typically appear only at a few patches, and most other patches are random noise or low-magnitude feature noises. More importantly, our noise parameters shall ensure that, the concept class is *not learnable by linear classifiers or constant degree polynomials*. We believe a (convolutional) neural network with ReLU-like activation is somewhat necessary.

Our final data distribution $\mathcal{D}$, and the training data set $\mathcal{Z}$ are formally given as follows.

**Definition 3.3** ($\mathcal{D}$ and $\mathcal{Z}$). *The distribution $\mathcal{D}$ consists of data from $\mathcal{D}_m$ w.p. $1 - \mu$ and from $\mathcal{D}_s$ w.p. $\mu$. We are given $N$ training samples from $\mathcal{D}$, and denote the training data set as $\mathcal{Z} = \mathcal{Z}_m \cup \mathcal{Z}_s$ where $\mathcal{Z}_m$ and $\mathcal{Z}_s$ respectively represent multi-view and single-view training data. We write $(X, y) \sim \mathcal{Z}$ as $(X, y)$ sampled uniformly at random from the empirical data set, and denote $N_s = |\mathcal{Z}_s|$. We again for simplicity focus on the setting when $\mu = \frac{1}{\text{poly}(k)}$ and we are given samples $N = k^{1.2}/\mu$ so each label $i$ appears at least $\widetilde{\Omega}(1)$ in $\mathcal{Z}_s$. Our result trivially applies to many other choices of $N$.*

### 3.2 LEARNER NETWORK

We consider a learner network using the following smoothed ReLU activation function $\widetilde{\text{ReLU}}$:

**Definition 3.4.** *For integer $q \geq 2$ and threshold $\varrho = \frac{1}{\text{polylog}(k)}$, the smoothed function $\widetilde{\text{ReLU}}(z) := 0$ for $z \leq 0$; $\widetilde{\text{ReLU}}(z) := \frac{z^q}{q\varrho^{q-1}}$ for $z \in [0, \varrho]$; and $\widetilde{\text{ReLU}}(z) := z - (1 - \frac{1}{q})\varrho$ for $z \geq \varrho$.*

Since $\widetilde{\text{ReLU}}$ is smooth we denote its gradient as $\widetilde{\text{ReLU}}'(z)$. We focus on $q = 4$ while our result applies to other constants $q \geq 3$ (see full version) or most other forms of smoothing.

---

[7]For instance, the marginal distribution of $Z = \sum_{p \in \mathcal{P}_v(X)} z_p$ can be uniform over $[1, 2]$.

[8]For instance, the marginal distribution of $Z = \sum_{p \in \mathcal{P}_v(X)} z_p$ can be uniform over $[0.2, 0.4]$.

The *learner network* $F(X) = (F_1(X), \dots, F_k(X)) \in \mathbb{R}^k$ is a two-layer convolutional network parameterized by $w_{i,r} \in \mathbb{R}^d$ for $i \in [k], r \in [m]$, satisfying

$$\forall i \in [k]: \quad F_i(X) = \sum_{r \in [m]} \sum_{p \in [P]} \widetilde{\mathsf{ReLU}}(\langle w_{i,r}, x_p \rangle)$$

Although there exists network with $m = 2$ that can classify the data correctly (e.g. $w_{i,r} = v_{i,r}$ for $r \in [2]$), in this paper, for efficient optimization purpose it is convenient to work on a moderate level of over-parameterization: $m \in [\mathsf{polylog}(k), k]$. Our lower bounds hold for any $m$ in this range and upper bounds hold even for small over-parameterization $m = \mathsf{polylog}(k)$.

**Training a single model.**   We learn the concept class (namely, the labeled data distribution) using gradient descent with learning rate $\eta > 0$, over the cross-entropy loss function $L$ using $N$ training data points $\mathcal{Z} = \{(X_i, y_i)\}_{i \in [N]}$. We denote the empirical loss as:

$$L(F) = \tfrac{1}{N} \sum_{i \in [N]} L(F; X_i, y_i) = \mathbb{E}_{(X,y) \sim \mathcal{Z}}[L(F; X, y)]$$

where $L(F; X, y) = -\log \frac{e^{F_y(X)}}{\sum_{j \in [k]} e^{F_j(X)}}$. We **randomly initialize** the network $F$ by letting each $w_{i,r}^{(0)} \sim \mathcal{N}(0, \sigma_0^2 I)$ for $\sigma_0^2 = 1/k$, which is the most standard initialization people use in practice.

To train a single model, at each iteration $t$ we update using gradient descent (GD):[9]

$$w_{i,r}^{(t+1)} \leftarrow w_{i,r}^{(t)} - \eta \, \mathbb{E}_{(X,y) \sim \mathcal{Z}} \, \nabla_{w_{i,r}} L(F^{(t)}; X, y) \tag{3.1}$$

We run the algorithm for $T = \mathsf{poly}(k)/\eta$ iterations. We use $F^{(t)}$ to denote the model $F$ with hidden weights $\{w_{i,r}^{(t)}\}$ at iteration $t$.

**Notations.**   We denote by $\mathbf{logit}_i(F, X) := \frac{e^{F_i(X)}}{\sum_{j \in [k]} e^{F_j(X)}}$. Using this, we can write down

$$\forall i \in [k], r \in [m]: \quad -\nabla_{w_{i,r}} L(F; X, y) = (\mathbb{1}_{i \neq y} - \mathbf{logit}_i(F, X)) \nabla_{w_{i,r}} F_i(X) \; .$$

# 4   MAIN THEOREMS AND EXPLANATIONS

We now state the main theorems (and the one for *self-distillation* is in the full paper).[10]

**Theorem 1** (single model). *For every sufficiently large $k > 0$, every $m \in [\mathsf{polylog}(k), k]$, every $\eta \leq \frac{1}{\mathsf{poly}(k)}$, suppose we train a single model using the gradient descent update (3.1) starting from the random initialization defined in Section 3.2, then after $T = \frac{\mathsf{poly}(k)}{\eta}$ many iterations, with probability $\geq 1 - e^{-\Omega(\log^2 k)}$, the model $F^{(T)}$ satisfies:*

- *(training is perfect): meaning for all $(X, y) \in \mathcal{Z}$, all $i \in [k] \setminus \{y\}$: $F_y^{(T)}(X) > F_i^{(T)}(X)$.*

- *(test accuracy is consistently bad): meaning that:*

$$\mathbf{Pr}_{(X,y) \sim \mathcal{D}}[\exists i \in [k] \setminus \{y\} : F_y^{(T)}(X) < F_i^{(T)}(X)] \in [0.49\mu, 0.51\mu] \; .$$

We shall give *technical intuitions* about why Theorem 1 holds in the full version. But, at a high-level, we shall construct a "lottery winning" set $\mathcal{M} \subseteq [k] \times [2]$ of cardinality $|\mathcal{M}| \in [k(1 - o(1)), k]$. It only depends on the random initialization of $F$. Then, with some effort we can prove that, for every $(i, \ell) \in \mathcal{M}$, at the end of the training $F^{(T)}$ will learn feature $v_{i,\ell}$ but not learn feature $v_{i,3-\ell}$. This means for those single-view data $(X, y)$ with $y = i$ and $\widehat{\ell}(X) = 3 - \ell$, the final network $F^{(T)}$ will predict its label wrong. This is why the final test accuracy is around $0.5\mu$.

Note the property that test accuracy consistently belongs to the range $[0.49\mu, 0.51\mu]$ should be reminiscent of message ⑤ in Figure 6, where multiple single models, although starting from different random initialization, in practice does have a relatively small variance in test accuracies.

---

[9]Our result also extends to the case when there is a weight decay, discussed in the full version.

[10]We shall restate these theorems in the full version with more details and a wider range of parameters.

**Ensemble.** Suppose $\{F^{[\ell]}\}_{\ell \in [K]}$ are $K = \widetilde{\Omega}(1)$ independently trained models of $F$ with $m = \mathsf{polylog}(k)$ for $T = O\big(\frac{\mathsf{poly}(k)}{\eta}\big)$ iterations (i.e., the same setting as Theorem 1 except we only need a small over-parameterization $m = \mathsf{polylog}(k)$). Let us define their ensemble

$$G(X) = \frac{\widetilde{\Theta}(1)}{K} \sum_\ell F^{[\ell]}(X) \tag{4.1}$$

**Theorem 2** (ensemble). *In the same setting as Theorem 1 except now we only need a small $m = \mathsf{polylog}(k)$, we have for the ensemble model $G$ in (4.1), with probability at least $1 - e^{-\Omega(\log^2 k)}$:*

- *(training is perfect): meaning for all $(X, y) \in \mathcal{Z}$, for all $i \in [k] \setminus \{y\}$: $G_y(X) > G_i(X)$.*

- *(test accuracy is almost perfect): meaning that:*

$$\mathbf{Pr}_{(X,y)\sim\mathcal{D}}[\exists i \in [k] \setminus \{y\} \colon G_y(X) < G_i(X)] \leq 0.001\mu \ .$$

As we discussed in Section 2.3, the reason Theorem 2 holds attributes to the fact that those lottery winning sets $\mathcal{M}$ depend on the random initialization of the networks; and therefore, when multiple models are put together, their "union" of $\mathcal{M}$ shall cover all possible features $\{v_{i,\ell}\}_{(i,\ell)\in[k]\times[2]}$. Moreover, our theorem only requires individual $K = \widetilde{\Omega}(1)$ models for ensemble, which is indeed "averaging the output of *a few* independently trained models".

### 4.1 Knowledge Distillation for Ensemble

We consider a knowledge distillation algorithm given the existing ensemble model $G$ (see (4.1)) as follows. For every label $i \in [k]$, let us define the truncated scaled logit as (for $\tau = \frac{1}{\log^2 k}$):

$$\mathbf{logit}_i^\tau(F, X) = \frac{e^{\min\{\tau^2 F_i(X), 1\}/\tau}}{\sum_{j\in[k]} e^{\min\{\tau^2 F_j(X), 1\}/\tau}} \tag{4.2}$$

(This should be reminiscent of the logit function with temperature used by the original knowledge distillation work (Hinton et al., 2015); we use truncation instead which is easier to analyze.)

Now, we train a new network $F$ from random initialization (where the randomness is independent of all of those used in $F^{[\ell]}$). At every iteration $t$, we update each weight $w_{i,r}$ by:

$$w_{i,r}^{(t+1)} = w_{i,r}^{(t)} - \eta\nabla_{w_{i,r}}L(F^{(t)}) - \eta' \mathbb{E}_{(X,y)\sim\mathcal{Z}} \left[ \left(\mathbf{logit}_i^\tau(F^{(t)}, X) - \mathbf{logit}_i^\tau(G, X)\right)^- \nabla_{w_{i,r}}F_i^{(t)}(X)\right] \tag{4.3}$$

**Notation.** Throughout the paper we denote by $[a]^+ = \max\{0, a\}$ and $[a]^- = \min\{0, a\}$.

This knowledge distillation method (4.3) is almost identical to the one used in the original work (Hinton et al., 2015), except we use a truncation during the training to make it more (theoretically) stable. Moreover, we update the distillation objective using a larger learning rate $\eta'$ comparing to $\eta$ of the cross-entropy objective. This is also consistent with the training schedule used in (Hinton et al., 2015).

Let $F^{(t)}$ be the resulting network obtained by (4.3) at iteration $t$. We have the following theorem:

**Theorem 3** (ensemble distillation). *Consider the distillation algorithm (4.3) in which $G$ is the ensemble model defined in (4.1). For every $k > 0$, for $m = \mathsf{polylog}(k)$, for every $\eta \leq \frac{1}{\mathsf{poly}(k)}$, setting $\eta' = \eta\mathsf{poly}(k)$, after $T = \frac{\mathsf{poly}(k)}{\eta}$ many iterations with probability at least $1 - e^{-\Omega(\log^2 k)}$, for at least 90% of the iterations $t \leq T$:*

- *(training is perfect): meaning for all $(X, y) \in \mathcal{Z}$, all $i \in [k] \setminus \{y\}$: $F_y^{(t)}(X) > F_i^{(t)}(X)$.*

- *(test accuracy is almost perfect): meaning that:*

$$\mathbf{Pr}_{(X,y)\sim\mathcal{D}}[\exists i \in [k] \setminus \{y\} \colon F_y^{(t)}(X) < F_i^{(t)}(X)] \leq 0.001\mu \ .$$

*Remark.* Theorem 3 necessarily means that the distilled model $F$ has learned all the features $\{v_{i,\ell}\}_{(i,\ell)\in[k]\times[2]}$ from the ensemble model $G$. This is consistent with our empirical findings in Figure 8: if one trains multiple individual models using knowledge distillation with different random seeds, then their ensemble gives no further performance boost.

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
