# OpenReview forum: "Towards Understanding Ensemble, Knowledge Distillation and Self-Distillation in Deep Learning"
_ICLR.cc/2023/Conference — ICLR 2023 notable top 5%_

### Official Review · Reviewer_ZnbC · 2022-10-15

**Confidence:** 2
**Correctness:** 3
**Technical Novelty And Significance:** 3
**Empirical Novelty And Significance:** Not applicable
**Recommendation:** 8

**Clarity, Quality, Novelty And Reproducibility:**

Overall, the paper is clearly written and well organized. As it was primarily a theory paper, it was fine to include few empirical results in the main paper. The multi-view data structure was novel to the community and made the first attempt to interpret how ensemble and knowledge distillation works in the field of deep learning. In addition, the theoretical analysis was rigorous except for some minor typos in Contradiction 2 in page 2, where $f_i$ and $g_i$ were used mixedly.

**Strength And Weaknesses:**

Strengths:
- The paper analyzed an underexplored problem of how ensemble works in deep learning methods. It pointed out contradictions when applying previous principles of traditional algorithms and shed light on understanding deep learning.
- For the theory side, the analysis was rigorous with clear definitions and concrete examples to help better digest the theorems.

Weaknesses:
- I am just curious whether the hypothesis of multi-view data structure can be extended to different data structures such as texts and graphs. Those data are very different from image data discussed in this paper and 'multi-view' might not be applicable to them.
- It is not a good practice to include "self-distillation" in the title but defer all details about it to the appendix, which was somewhat misleading even though it was due to page limit.

**Summary Of The Paper:**

This paper investigated the superior performance of ensemble, knowledge distillation and self-distillation in the field of deep learning. As previous principles of ensemble on traditional machine learning algorithms do not apply to deep learning methods, a novel perspective of "multi-view" data was proposed to explain the phenomenon. Both empirical and theoretical analysis support multi-view theory for ensemble in deep learning.

**Summary Of The Review:**

The paper made an attempt to explain the superior performance of ensemble, knowledge distillation and self-distillation with the theory of "multi-view" data structure and provided insights on understanding those techniques in deep learning. The analysis is rigorous and thus I tend to accept it.

---

> ### Author Response · Authors · 2022-11-19
> **Response to Reviewer ZnbC**
>
> We sincerely thank the reviewer for carefully reading our paper and pointing out our typos. We are very grateful that reviewer agreed that our work is a “first attempt to interpret how ensemble and knowledge distillation works in the field of deep learning”
>
> ---
>
> Q: I am just curious whether the hypothesis of multi-view data structure can be extended to different data structures such as texts and graphs. Those data are very different from image data discussed in this paper and 'multi-view' might not be applicable to them.
>
> A: Great question. Yes, we know that ensemble/distillation works in many data types in practice. For instance for language models, arxiv paper 2110.04260 is a very modern, smart way to perform self-distillation (over randomly chosen experts). We chose to analyze image data because it is more theory friendly and easy to visualize. We can perhaps say a few words about language models (as we are not domain experts on other domains).
>
> In a masked language modeling task, one is to predict some masked words (out of a vocabulary size of, say 50257 for GPT2). It is a 50257-class classification task. Imagine if the sentence is “Alice is pretty [MASK] about the review scores” and the correct label for [MASK] is “excited”. Then, it is quite plausible that the soft label is 0.8 x “excited” + 0.1 x “happy” + 0.01 x “Alice” + 0.01 x “review” + and so on. Those hidden soft labels may force the model to learn some hidden features that otherwise are not necessary to predict the hard label. In particular, this 0.1 x “happy“ shall encourage the model to learn the sentiment as opposed to merely looking at the grammar (recall “excited about” is a common phrase). This 0.01 x “Alice” shall encourage the model to learn the fact that it is still Alice that is excited, as opposed to forgetting about the subject. Thus, we believe for text data, there is also some form of “multi-view” structure but it can be hard to analyze in theory.
>
> We also gave a multi-view example in the context of Ads model (binary classification) in our response to Reviewer qyvd. Feel free to take a look there: https://openreview.net/forum?id=Uuf2q9TfXGA&noteId=LRDBu4Hlfy.
>
> ---
>
> Q: It is not a good practice to include "self-distillation" in the title but defer all details about it to the appendix, which was somewhat misleading even though it was due to page limit.
>
> A: We thank the reviewer for making a good point. If you allow us to explain self-distillation in one sentence: it is performing an implicit ensemble of two models (teacher + student) and knowledge distillation (to distill it to the student model). We quickly mentioned this on page 2 but yes it deserves more explanations. If we get an additional page during camera ready, we can perhaps add it in.
>
> ---
>
> Q: We noticed the reviewer chooses to rate our paper as “Aspects of the contributions exist in prior work”
>
> A: Well, we’d like to ask if the reviewer can re-think a bit if we qualify as “The contributions do not exist in prior works.”
>
> First, although there are prior works (including boosting, bootstrapping) studying how ensemble works, as we cited on pages 1-2, they consider the setting when the individual models are different. This is not the case for deep learning. In deep learning, even ensemble of individual models with the same architecture and same size, and using the same training data, can greatly improve test accuracy. We studied this hard setting where all the models are exactly identical except for random seeds. This was thought to be a mystery in deep learning for very long.
>
> Second, to the best of our knowledge, recent theories in deep learning are also not applicable to resolve such mysteries. In particular, we have elaborated why those kernel/NTK approaches give even contradictory results to what happens in practical deep learning (Section 2.1). We also explained why theories that study neural networks with Gaussian-like inputs, or those that relate ensemble performance simply to the sub-optimality of non-convex optimization fail to address such mysteries in practice (Section 2.2).
>
> We hope this would be sufficient to support the claim that “our contributions do not exist in prior works.” Thanks for your reconsideration!

---

### Official Review · Reviewer_GAwQ · 2022-10-24

**Confidence:** 4
**Clarity, Quality, Novelty And Reproducibility:** All aspects are good.
**Correctness:** 4
**Technical Novelty And Significance:** 4
**Empirical Novelty And Significance:** 4
**Recommendation:** 10

**Strength And Weaknesses:**

# Strength
- The writing and presentation are good. It is a pleasure to read this paper.
- The empirical results on neural tangent features and deep neural networks are novel and interesting. The authors then provide thorough explanations for them, which is appreciated.
- The theoretical results are rigorous.

# Weaknesses
- We know new theories usually motivate new algorithms. I want to know if there are some potential future works to better exploit these theories.

**Summary Of The Paper:**

The paper studies the theories of ensemble and knowledge distillation. Insightful studies regarding NTKs and deep neural networks are performed and novel theoretical results based on the multi-view structure of data are established. I get many new insights after reading this paper.

**Summary Of The Review:**

This is an outstanding paper in multiple aspects, so I suggest acceptance.

---

> ### Author Response · Authors · 2022-11-19
> **Response to Reviewer GAwQ**
>
> We sincerely thank the reviewer for the time reading our paper, and liking our presentations.
>
> We especially want to thank the reviewer for noticing that “*empirical results* on neural tangent features and deep neural networks are novel and interesting”
>
> Yes, we feel that not only our theory is new, our empirical findings on (the failure of) using NTK features to study ensemble/distillation may also be new. We did not see this being recorded, and among experts we discussed offline, we noticed some of them indeed thought ensemble works because having more models could create a larger feature space (in the language of NTK) — but as we have experimentally demonstrated in this paper, such intuitions may be wrong for that of deep learning.
>
> In case you're interested, we also performed an experiment in Figure 8 in the appendix, to show that ensemble over multiple runs of knowledge distillations gives no better performance. This further supports our theory. (For instance, it means after distillation the view features are already fully learned, so performing additional ensemble is useless). We think this experiment is also new and very cute.
>
> Last but not least, we thank your question on asking for whether our new theory implies new algorithms.  We found this an excellent question, and *answered it elsewhere* to a wider audience: https://openreview.net/forum?id=Uuf2q9TfXGA&noteId=z59ELY2gS5C.

---

### Official Review · Reviewer_qyvd · 2022-10-24

**Confidence:** 4
**Correctness:** 4
**Technical Novelty And Significance:** 4
**Empirical Novelty And Significance:** 4
**Recommendation:** 10

**Clarity, Quality, Novelty And Reproducibility:**

Clarity: Good
Quality: Good
Novelty: Godd

**Strength And Weaknesses:**

Strength:

1. The novelty of the paper is quite strong. I believe the such analysis is new and sound.

2. There is quite a good match between theory and practice, especially the construction of the multiview dataset is an exciting and meaningful proxy for the practical case while being theory-friendly.

3. The writing of the main context is well organized. It is easy for the reader to get to know the most critical intuition of the theory.

4. I like the experiment used in this paper. It is concise but well supports the theory.

Weakness:

1. I am not able to open the supplementary of the paper but reviewing this paper, reading the appendix is very necessary and thus my reading of the appendix is via the arxiv. The structure of the appendix is good but I feel a more simplified overall technical/intuition is desirable.

2. To obtain such a theory, there are still a lot of 'artificial' in the problem setup such as the use of smoothed ReLU, and specific restrictions of the data distribution. Those slightly increase the gap between theory and practice. But this point does not downgrade my point, understanding deep learning in theory is very hard and this paper has done excellent work toward that.


**Summary Of The Paper:**

This paper studies the theoretical explanation of why the ensemble and knowledge distillation works for the deep learning model. It shows the different behavior of classic parameterization of neural networks and the random feature model (e.g. NTK). It then explains via the idea of multi-view data that the network infers via generalizable learning of a subset of features and memorizing another subset of data. Such analysis differs from the classic analysis that decomposes optimization and generalization.

Overall, I find the insight of the paper quite interesting, and such intuition is well supported by the empirical evidence using the practical network. The construction of the multi-view data for analysis is closely related to the practical settings.


**Summary Of The Review:**

Questions:

1. Almost all the result suggests a 100% training accuracy but this is different from almost 0 cross-entropy loss. I feel all the results will not hold if the model is large and trained with almost zero cross-entropy loss? At least, in this case, the distillation via logits seems to be no more useful. Pls correct me if I am wrong.

2. Following 1, how do the results look like if we consider regression rather than classification?

3. Following 1 and 2, I was curious if we do distillation on the latent feature space rather than logit space, will the result become very different? From the feature learning perspective, distill on the feature space seems more reasonable?

---

> ### Author Response · Authors · 2022-11-19
> **Response to Reviewer qyvd**
>
> We sincerely thank the reviewer for the time reading our paper and finding our supplementary materials on arxiv. We apologize for any inconvenience, and we have re-uploaded a zip file that contains the same supplementary materials. (The earlier submitted version can be opened by linux unzip, 7-Zip, and WinRAR.)
>
> ----
>
> Q1: Almost all the result suggests a 100% training accuracy but this is different from almost 0 cross-entropy loss. I feel all the results will not hold if the model is large and trained with almost zero cross-entropy loss? At least, in this case, the distillation via logits seems to be no more useful. Pls correct me if I am wrong.
>
> A1: Sorry that we did not explain this perfectly. On one hand, you’re right and during training, if there is *no* weight decay, single models will learn to have almost zero cross-entropy loss, but this will be only at a polynomial speed — see full statement of Thm 1 on page 52. Specifically, after poly(k) steps, it satisfies for any wrong label i!=y, the output $F_y(x) > F_i(x) + \Omega(\log k)$ — which implies xE loss is around 1/poly(k). This is okay, as distillation will use the logits $F_i(x)$ as teachers.
>
> Remarks: our stated versions of the theorem (that distillation works) will stop working if single models are trained for exp(k) time.  But if one is willing to run distillation for exponential time as well, then it will work again. Furthermore, as we briefly mentioned in Footnote 9, our result also holds if there is weight decay --- in that case the cross entropy loss will not go to zero.
>
> ---
>
> Q2: Following Q1, how do the results look like if we consider regression rather than classification?
>
> A2: See our A3 below and we answer the questions together.
>
> ----
>
> Q3: Following Q1 and Q2, I was curious if we do distillation on the latent feature space rather than logit space, will the result become very different? From the feature learning perspective, distill on the feature space seems more reasonable?
>
> A3: We first answer Q3 and then Q2. It is known that ensemble/distillation also works on latent feature space. This is especially helpful if one only has a binary classification task (like the Ads model for click vs not-to-click). The use of latent space distillation may not be needed for language models (as the classification vocab size is e.g. 50k for GPT2).
>
> As for how our theory fits into this, take binary classification as an example. A naïve way to translate our theory to this case, is to consider binary classification on top of our k classes, e.g. k/2 of the classes are labeled 0 and the others are label 1. In this case, the original signal of k classes are now 'latent', and our theorem can extend to this case.
>
> More broadly, in practice, latent space right before the last output layer is usually thought to be a most informative feature representation of the data. (It can be used for downstream tasks, etc.) If one views those “latent space features” as “classes”, then the data may still be “multi-view” in them.  [[[ For instance, consider Ads models, the hidden latent space features may have different meanings: they may correspond to the user’s past browser history, user’s geo info, user’s purchasing history, etc. Now, the user may eventually click this ads for “multiple” reasons, and some reasons might even be missing from the data (imagine if the user disabled Cookies, etc).  In this example, our theory of the “multi-view” data for distillation/ensemble would still apply. ]]
>
> Finally, back to Q2, we believe for regression it is similar. The latent feature space right before the last output layer, may serve as “feature representations” and they may correspond to different views of the data even on a regression task.
>
> ---
>
> Q4: A more simplified overall technical/intuition is desirable.
>
> A4: We thank the reviewer for this suggestion. We’d like to hear what you think. Currently, we have a short explanation of the intuitions in Section 2.3 on page 4-5; and in the appendix, after introducing notations, we have a 1.5-paged Section G on page 27 (or Section B on page 24 in the arxiv) sketching the main reason for what makes single models perform “bad” on multi-view data. We cited it on Page 8, and its purpose is to explain technically what Section 2.3 means.
>
> We agree that perhaps we should make the intuitions earlier, but currently we see no way to bring them all to the first 9 pages. One way to do so is to add a crisp version of that during camera ready (perhaps we shall have one additional page?) But we are debating this with moving more figures / experiments to the first 9 pages. What would you suggest?
>
> Thanks again for supporting our work!

---

### Official Review · Reviewer_Y3qu · 2022-10-24

**Confidence:** 4
**Correctness:** 3
**Technical Novelty And Significance:** 3
**Empirical Novelty And Significance:** 3
**Recommendation:** 8

**Clarity, Quality, Novelty And Reproducibility:**

The idea of "multi-view" is interesting, and provides an intuitive explanation for various observations in distillation.

**Strength And Weaknesses:**

Strength:

1. The authors present theoretical results showing that a single model is guaranteed to have 0 training error while having a high testing error with high probability; the ensemble can provably improve the testing accuracy.

2. The author shows that the ensemble can be efficiently distilled into a single model. This understanding is fundamentally different from the standard NTK settings.

3. The idea of "multi-view data" is intuitive and provides a natural and convincing explanation for various empirical observations for distillation.

Questions and weaknesses:

1. One major concern is that the supplementary material cannot be opened due to a file format error, and I'm not able to see the results of self-distillation and empirical evaluation, which seem to be important aspects of this work.

2. In the example of Section 2.3 and for a single model, why the network can only pick up one of the features in $v_1, v_2$ with the 90% of the data, and only memorize the remaining 10% of the data, instead of learning from the other relevant feature in $v_1, v_2$?



**Summary Of The Paper:**

This paper takes a step toward understanding ensemble and knowledge distillation. The authors consider the challenging setting where the teacher model is an average of several models of the same structure, or even the teacher has an identical structure as the student model. The authors developed a theory that the distillation from several independently trained neural networks can improve the performance when the data has a "multi-view" structure.

**Summary Of The Review:**

This paper presents an interesting and intuitive explanation for distillation. However, one major concern is that I'm not able to open the supplementary material due to some file format error. I'm not able to fully assess this paper.

---

> ### Author Response · Authors · 2022-11-19
> **Response to reviewer Y3qu**
>
> We sincerely thank the reviewer for the time reading our paper, and liking our concept of the "multi-view" structure and found our explanation convincing.
>
> Q: Supp materials?
>
> A: We apologize for the inconvenience. Upon close investigation we found our original zip file can be opened by softwares like linux unzip, 7-Zip or WinRAR but not by some others. We have re-attached the same supplementary material and hope this time it works for all softwares.
>
> In particular, we had a Section C (starting on page 15) to summarize our empirical findings.
> 1. Our Figs 4+5+9 try to justify the multi-view hypothesis
> 2. Our Figs 6 explain why the popular neural kernel techniques fail to explain ensemble/distillation for deep learning. (In fact, we even believe this experiment is new.)
> 3. Fig 7 shows ensemble/distillation do not work over Gaussian-like data, so some special structural assumption (like our multi-view one) may be necessary.
> 4. Fig 8 shows an interesting experiment that ensemble over knowledge distillation gives no better performance. This further supports our theory. (For instance, it means after distillation the view features are already fully learned, so performing additional ensemble is useless).
>
> ----
>
> Q: In the example of Section 2.3 and for a single model, why the network can only pick up one of the features in v1,v2 with the 90% of the data, and only memorize the remaining 10% of the data, instead of learning from the other relevant feature in v1, v2?
>
> A: Sorry that we did not explain this perfectly. Depending on the initial randomness, a neural network may learn some features faster than others (say, v1 faster than v2). The multi-view structure ensures that as long as v1 is learned, then the 90% multi-view data (plus 5% single-view using v1) can be classified perfectly; because of this, those data will contribute **negligibly** to the cross-entropy loss. (Their gradients close to zero.) Now, to continue to learn v2, the neural network can only rely on those 5% single-view data. We designed the setting to put the data size in a sweet spot, so that those 5% single-view data is **not enough** (i.e., too few) to learn v2, and thus NN can only memorize them (recall NN will always give perfect accuracy on training data).
>
> (Of course, in practice, depending on the data size, it may allow some features to be learnable and some others not. We analyzed an ideal case — of two features, one for each case — for simplicity.)
>
> --------
>
> Q: We noticed the reviewer chooses to rate our paper as “Aspects of the contributions exist in prior work”
>
> A: Well, we’d like to ask if the reviewer can re-think a bit if we qualify as “The contributions do not exist in prior works.”
>
> First, although there are prior works (including boosting, bootstrapping) studying how ensemble works, as we cited on pages 1-2, they consider the setting when the individual models are different. This is not the case for deep learning. In deep learning, even ensemble of individual models with the *same* architecture and *same* size, and using the *same* training data, can greatly improve test accuracy. We studied this hard setting where all the models are exactly identical except for random seeds. This was thought to be a mystery in deep learning for very long.
>
> Second, to the best of our knowledge, recent theories in deep learning are also not applicable to resolve such mysteries. In particular, we have elaborated why those kernel/NTK approaches give even contradictory results to what happens in *practical* deep learning (Section 2.1). We also explained why theories that study neural networks with Gaussian-like inputs, or those that relate ensemble performance simply to the sub-optimality of non-convex optimization fail to address such mysteries in practice (Section 2.2).
>
> We hope this would be sufficient to support the claim that “our contributions do not exist in prior works.” Thanks for your reconsideration!

---

> > ### Comment · Reviewer_Y3qu · 2022-12-06
> > **Thank you for your response.**
> >
> > Thank you for the detailed response. It resolves all my concerns. In particular, I found it very interesting that the effectiveness of self-distillation can also be explained from the multi-view perspective. The empirical results and theoretical insights are nicely presented in the appendix. I have raised my score.

---

### Author Response · Authors · 2022-11-05
**Supplementary zip file re-uploaded**

Dear reviewers,

It was brought to our attention that our supplementary material zip file (containing a single pdf) cannot be opened on some computers. That's very weird, as it works well on our end...

We have now used a different zip software and re-uploaded the same supplementary material. Please kindly let us know if you still have issues opening the file?

We will provide detailed response to your review comments later. Thanks a lot for your time and efforts! Really appreciate that.

Best,
Authors

---

### Author Response · Authors · 2022-11-19
**Can our work motivate new algorithms? Yes.**

We noticed that Reviewer GAwQ asked us if there could be potential future (practical) works to better exploit our theories. We find it an excellent question and would like to answer it here to a wider audience.

We give two applications below. (If we have an opportunity to present this work at ICLR we can perhaps elaborate more.)

---

One message of our paper is that, the performance gain "obtained via distillation / ensemble", is very much **orthogonal to** that "obtained by increasing the model size”. It is in fact a misconception that “ensemble works because the model is larger”. [[ Recall we explained on page 1 (later with experiments) that training an average of 10 models doesn’t give significant accuracy gain although the model is huge; while self distillation works and gives a performance boost without increasing model size. Practitioners sometimes concatenate network units together and call it ensemble — but it is fundamentally different from what we mean in the paper. ]]

Therefore, in nearly all the deep learning applications where ensemble/distillation is not fully exploited – regardless of how large/complex the model is, we believe there is still room for improvement to do ensemble/distillation. We are working on an ads model to take this into account, and thanks to (ideas similar to) self distillation, we are able to keep the model size as small as individual models.

----

Our paper can also give advises on **Model Compression in practice**. Nowadays, for all good reasons, it is desirable to compress teacher models (like GPT3) to smaller sizes, while (hopefully) maintaining its good performance at least on certain downstream tasks. Model compression is usually done in a similar way to self knowledge distillation (to match the latent space or soft labels).

If one compares this to pure self-distillation --- that is, given a teacher model, to train a student model of the *same size* to distill from the teacher's model --- there is already a performance boost. Therefore, it is not surprising if one trains a teacher WideResNet of width 320, and then trains student WideResNet of width 160 or even 80 to distill from the teacher, then the student model can *match the teacher's performance* (we verified it in practice). Can we say therefore Model Compression works? **NO.** In this case, the matching performance is primarily because distillation encourages the student model to learn “more view features", while the teacher --- although being larger --- may still be missing some view features.

In contrast, suppose the teacher is a WideResNet of width 320 but already trained using distillation --- say from an ensemble of ten width-320 models. In this case, to use a smaller student model of width 160 to learn this teacher, it can hardly match its performance. (One may want to compare this statement to our experiments in Figure 8; they are in the same spirit.)

In sum, it is perhaps not surprising that people report (sometimes) contradictory experimental results in the context of model compressions – sometimes it works, sometimes it doesn’t. Our theory suggests that it largely depends on how the teacher model was trained. We believe following this line of thoughts, there could be other more systematic ways to perform model compression. For instance, our framework immediately suggests that instead of looking at the teacher's test performance, one should measure its feature diversity to judge whether it is good or not. We are working on adding such a regularizer in practice to obtain a better teacher.

---

### Comment · Program_Chairs · 2023-03-21
**Congratulations on winning the ICLR 2023 Outstanding Paper Honorable Mention!!**

Hello authors,

Big congratulations! For more details about the award process this year, please see https://blog.iclr.cc/2023/03/21/announcing-the-iclr-2023-outstanding-paper-award-recipients/

Best,

Mengdi, on behalf of ICLR 2023 PCs

---

### Decision · Program_Chairs · 2023-01-20

**Decision:**

Accept: notable-top-5%

**Justification For Why Not Higher Score:**

N/A

**Justification For Why Not Lower Score:**

See above.

**Metareview: Summary, Strengths And Weaknesses:**

This work analyzes the case of why ensembles of neural networks, each member differing only by random initialization, can somehow improve generalization, i.e., held-out test set accuracy. They provide theory showing that "multi-view" structures in data leads to ensembles provably improving test accuracy, and moreover, such ensembles can also be distilled into a single model.

Reviewers all praised the theoretical analysis, particularly with its unique perspective that differs from the popular one of neural tangent kernels. The assumptions and claims were rigorous and well-founded. Quoting a reviewer: "The idea of "multi-view data" is intuitive and provides a natural and convincing explanation for various empirical observations for distillation." This paper has quite a good match between theory and practice, albeit with some simplifying assumptions such as a smoothed ReLU and on the data distribution.

All reviewers had quite high ratings (highest among my stack as well), and I agree with their consensus.

**Note From Pc:**

if the above contains the word "oral" or "spotlight" please see: "oral" presentation means -> notable-top-5% and "spotlight" means -> notable-top-25%. As stated in our emails, we are disassociating presentation type from AC recommendations